# A nanoscale photonic thermal transistor for sub-second heat flow switching

Ju Won Lim [1,5], Ayan Majumder [2,5], Rohith Mittapally [2], Audrey-Rose Gutierrez [3], Yuxuan Luan [2], Edgar Meyhofer [2,4] ✉ & Pramod Reddy [1,2,3] ✉

Control of heat flow is critical for thermal logic devices and thermal management and has been explored theoretically. However, experimental progress on active control of heat flow has been limited. Here, we describe a nanoscale radiative thermal transistor that comprises of a hot source and a cold drain (both are ~250 nm-thick silicon nitride membranes), which are analogous to the source and drain electrodes of a transistor. The source and drain are in close proximity to a vanadium oxide ($VO_x$)-based planar gate electrode, whose dielectric properties can be adjusted by changing its temperature. We demonstrate that when the gate is located close ( < ~1 μm) to the source-drain device and undergoes a metal-insulator transition, the radiative heat transfer between the source and drain can be changed by a factor of three. More importantly, our nanomembrane-based thermal transistor features fast switching times ( ~ 500 ms as opposed to minutes for past three-terminal thermal transistors) due to its small thermal mass. Our experiments are supported by detailed calculations that highlight the mechanism of thermal modulation. We anticipate that the advances reported here will open new opportunities for designing thermal circuits or thermal logic devices for advanced thermal management.

Thermal radiation plays an important role in heat-to-electricity conversion[1–3], near-field heat transfer[4–7], and photonic cooling[8–10]. Further, theoretical work[11–17] has also explored how radiative thermal transistors can be created. Analogous to electrical transistors, which are three-terminal transconductance devices where the current flow between the source and drain is regulated by a gate voltage[18,19], thermal transistors are based on three-terminal thermal designs where the temperature of a gate electrode controls the heat flow between two other terminals (source and drain), enabling heat flow switching. More recently, several theoretical works have also explored the implementation of thermal transistors via quantum effects[20–23]. While some recent experimental studies[24–27] have explored the creation of thermal

transistors based on heat conduction and radiation, so far no near-field photonic thermal transistors—i.e., devices that control the flow of photons based on temperature or thermal changes—have been experimentally explored. Furthermore, these past demonstrations feature relatively slow switching times due to the large thermal mass of the employed devices and the challenges in rapidly modulating device material properties. The experimental demonstration of fast thermal transistors has also been hampered by the lack of the required experimental apparatus for precise thermal measurements and the challenges associated with the fabrication of appropriate micro- and nanoscale devices. Given the substantial potential of photonic thermal transistors for thermal energy utilization and modulation, along with

[1]Department of Materials Science and Engineering, University of Michigan, Ann Arbor, MI, USA. [2]Department of Mechanical Engineering, University of Michigan, Ann Arbor, MI, USA. [3]Department of Electrical Engineering and Computer Science, University of Michigan, Ann Arbor, MI, USA. [4]Department of Biomedical Engineering, University of Michigan, Ann Arbor, MI, USA. [5]These authors contributed equally: Ju Won Lim, Ayan Majumder. ✉e-mail: meyhofer@umich.edu; pramodr@umich.edu

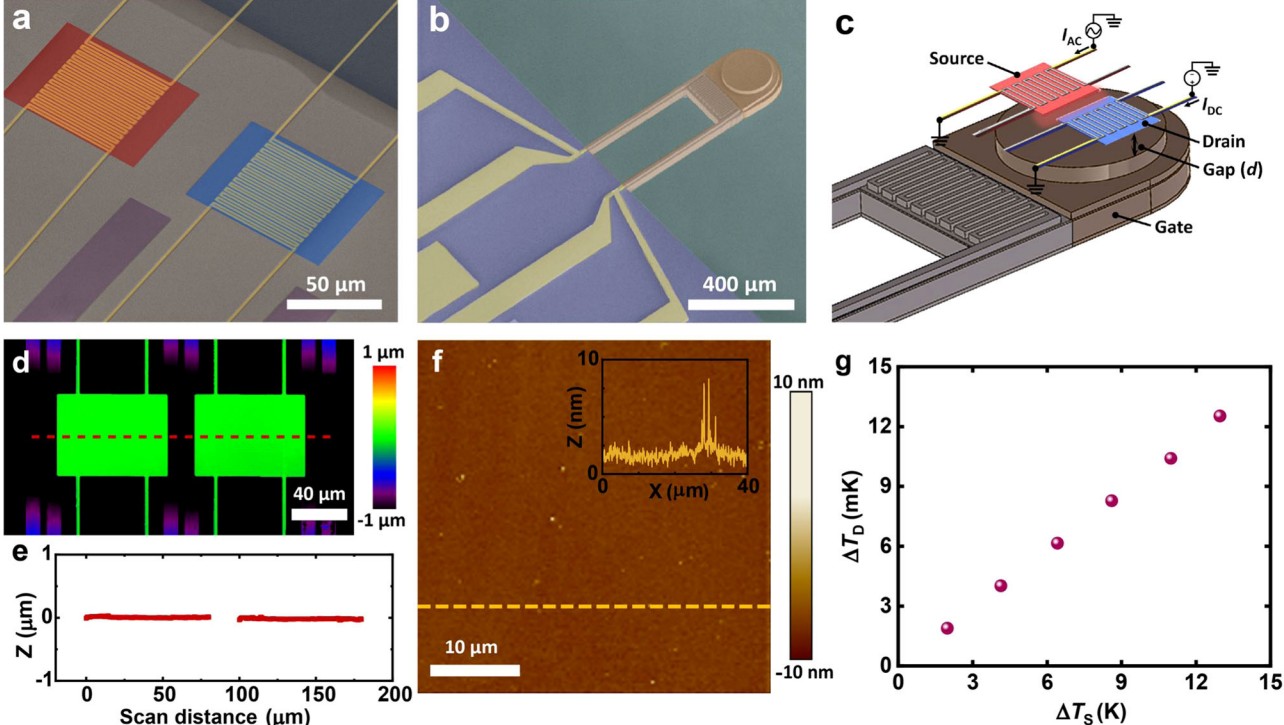

**Fig. 1 | Schematic of the experimental configuration and device characterization. a** False-colored scanning electron microscope (SEM) image of the nanofabricated source and drain device, showing a coplanar silicon nitride (SiN) membrane device with integrated Pt serpentine. **b** False-colored SEM image of the fabricated gate device featuring an incorporated Si heater and a layer of VO$_x$. **c** Schematic of the nanofabricated three-body system. $I_{AC}$ represents the amplitude of the AC current supplied to the source and $I_{DC}$ is the magnitude of the DC current supplied to the drain. **d** Confocal microscope scan of the source-drain device. **e** Profile along the dashed scan line in Fig. 1d, demonstrating excellent planarity of the device. **f** Atomic force microscopy (AFM) image of the gate coated with VO$_x$. The inset in Fig. 1f indicates a scanned height profile along the dashed line. **g** The amplitude of the temperature oscillation of the drain ($\Delta T_D$) was studied as a function of the amplitude of temperature oscillations of the source ($\Delta T_S$). The source temperature was modulated at a frequency of 2 Hz, while the gate positioned far away ($d \sim 25\,\mu$m) from the source-drain device, while maintaining a temperature of 25 °C.

its applicability in thermal logic circuits and computing, the necessity for an in-depth inquiry into this technology is crucial[11,24,26,28].

In this work, we experimentally demonstrate a nanoscale radiative thermal transistor—a three-terminal system consisting of source, drain, and gate devices (see Fig. 1a, b). We show that the heat current between the source and the drain ($\dot{Q}_{S-D}$) can be controlled by changing the temperature of the gate, which is located below the source-drain device at a gap size $d$ (see Fig. 1c) and is coated with a phase change material (vanadium oxide, VO$_x$) whose dielectric properties are strongly dependent on temperature[29,30]. We note that VO$_x$ was selected as the gate material because it undergoes a metal-insulator transition at a relatively low temperature (~ 340 K) compared to other phase-transition materials (e.g., Ti$_3$O$_5$, LaCoO$_3$, Ti$_2$O$_3$, NbO$_2$), whose transition temperatures range from 448 to 1081 K[31]. The relatively low transition temperature makes it more suitable for practical applications operating in ambient conditions. Using these microfabricated devices, we show that the heat transfer between the source and the drain (see Fig. 1a and Supplementary Fig. 1) can be modulated by up to a factor of three when the gap size between the source-drain device and the gate is less than ~ 1 μm and the gate undergoes a thermally-driven metal-insulator transition. Importantly, we demonstrate that our radiative thermal transistor achieves a switching time of less than a second (~ 500 ms), which is ~ 200 times faster than the fastest reported three-terminal thermal transistors demonstrated to date (1.7 min)[25] and even faster than the switching times in older works, which ranged from a few minutes to an hour[24–26]. This sub-second scale fast-switching thermal transistor offers significant potential to explore new avenues in thermal-based computing systems, signal processing, and data transmission[28,32,33].

## Results

In this work, we employed two independently microfabricated devices. The first device (see Fig. 1a), referred to as the source-drain device, consists of two SiN membranes that form the source (i.e., the thermal emitter) and drain (i.e., the thermal receiver) of our thermal transistor and are coplanar. These membranes are 250 nm thick and contain a serpentine platinum resistor. This serpentine line functions as a heater in the source and as a thermometer in the drain (Fig. 1a). The source and the drain are separated by a fixed distance of 20 μm and are suspended by long beams affixed to a silicon handler chip. The gap size of 20 μm was intentionally chosen to ensure that the near-field radiative heat transfer effect, which becomes considerable when the gap size is smaller than the thermal wavelength $\lambda_{th}$ (~ 10 μm at 300 K), is negligible[34,35]. We note that the source and drain membranes feature internal tensile stresses that ensure that they have excellent planarity and are co-planar, as confirmed by laser scanning confocal microscopy (see Fig. 1d, e). Shields designed to attenuate heat exchange between beams are also visible in the confocal microscopy image. Further details about the fabrication of the source-drain device are provided in Supplementary Note 1.

The bottom device used in this work called the gate (see Fig. 1b), consists of the suspended portion of a Si device that is supported by two beams. The suspended region features a doped silicon resistor heater/thermometer and a 15 μm-tall silicon mesa, which is coated with a 150 nm-thick VO$_x$ film. To achieve the phase transition of VO$_x$ (VO$_x$ films undergo a metal-insulator transition at ~ 68 °C[36,37]), we applied a bipolar voltage to the integrated silicon resistor to heat the gate. The bipolar voltage helped maintain the voltage of the gate region close to 0 V and minimized the electrostatic interaction between the top and

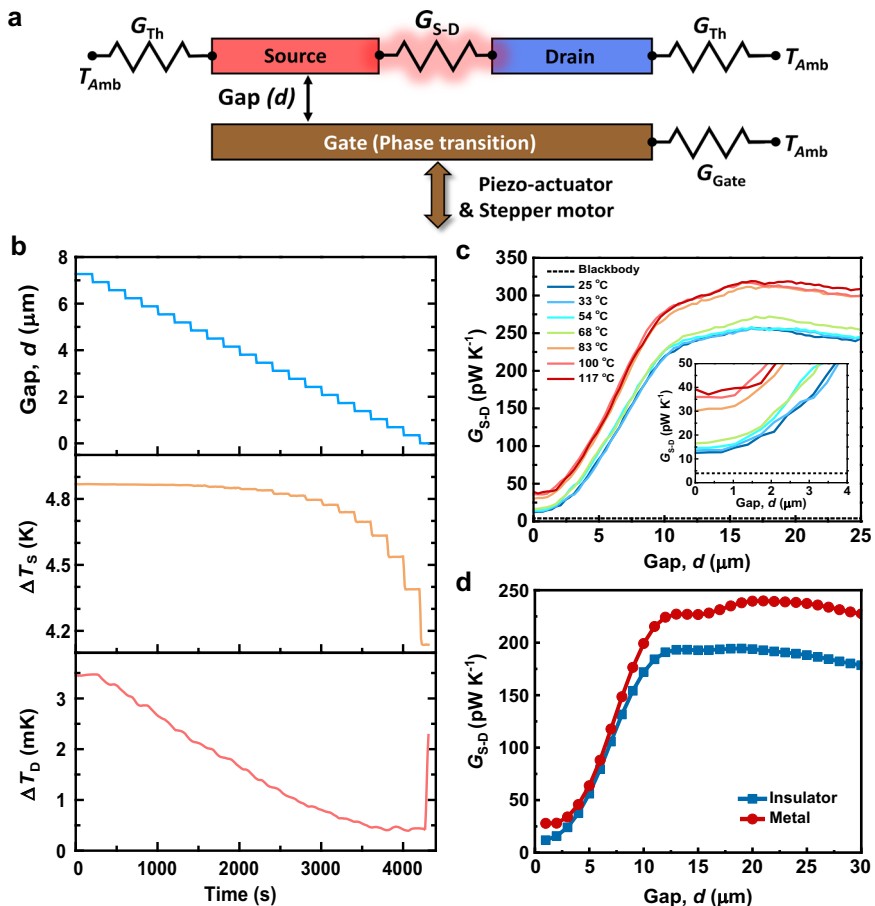

**Fig. 2 | Measured and computationally predicted conductance between the source and the drain device. a** Thermal resistance network corresponding to the thermal transistor. Both the suspended source and drain membranes have a total thermal conductance of $G_{Th}$, while the gate has a beam conductance of $G_{Gate}$. The thermal conductance between the source and the drain is represented by $G_{S-D}$. The height of the gate is adjusted using a piezoelectric actuator and a stepper motor, separated from the source-drain device by a gap distance $d$. Devices are placed in an environment at ambient temperature, $T_{Amb}$. **b** Time series of a representative measurement as the gate approaches the source-drain device at 25 °C. The top graph shows the measured gap distance ($d$) between the source-drain device and the gate until contact. The middle and bottom panels show the amplitude of temperature oscillations for the source ($\Delta T_S$) and the drain ($\Delta T_D$), respectively, as a function of time. **c** Experimental data for $G_{S-D}$ as a function of the gap size ($d$). The effect of the phase transition can be clearly seen when the temperature of the gate changes from 25 °C to 117 °C. A noticeable change in $G_{S-D}$ was observed when the gate temperature crossed the phase transition threshold (from 68 °C to 83 °C). **d** SCUFF-EM calculations of $G_{S-D}$ when the gate is in the insulator and metallic phases. It can be clearly seen that there is a good qualitative agreement with the experimental data in panel (**c**).

bottom devices. As described in the Methods section, the silicon resistor is used to heat up the temperature of the gate region, which enabled us to systematically vary the temperature of the gate from 298 to 390 K to ensure a complete phase transition and thus control the phase (metal or insulator) of the $VO_x$ film integrated into the gate. More details about the fabrication and characterization of the gate device can be found in Supplementary Note 2, Supplementary Note 3, and Supplementary Fig. 6.

In order to demonstrate modulation of heat flows via thermal inputs to the gate, we first placed the two devices (both top and bottom devices) in a custom-built nanopositioner[38] that allows parallelization of the devices as well as control of the gap size ($d$) between them in a high vacuum environment (~1 μTorr). In Fig. 1c, we schematically depict the relative orientation of the devices. Once the system thermally equilibrated with the vacuum chamber (i.e., when the temperature drift of the devices was <1 mK/hr), we first adjusted the gap size ($d$) to 25 μm. Next, we applied an alternating electrical current (AC) of suitably chosen amplitude ($I_{AC}$) at a frequency ($f$) of 1 Hz to the PR heater of the source to sinusoidally modulate the temperature of the source ($\Delta T_S$) at 2 Hz. The radiative heat current from the source to the drain produced temperature oscillations of the drain ($\Delta T_D$) at 2 Hz,

which were quantified by passing a fixed sensing DC current ($I_{DC}$) of 10 μA through the drain's PR thermometer to measure the voltage fluctuations at 2 Hz using a lock-in amplifier (see "Methods"). We note that there are negligible thermal gradients within the suspended membranes (discussed in more detail in Supplementary Fig. 7d), which is key to accurately measuring the heat fluxes. The measured $\Delta T_D$ as a function of $\Delta T_S$ is shown in Fig. 1g at a large separation distance ($d = 25$ μm) with a gate temperature ($T_G$) of 25 °C and is found to be linearly related. These data enabled the calculation of the thermal conductance ($G_{S-D}$) between the source and drain devices via $G_{S-D} = (G_{Th}) \times [(\Delta T_S / \Delta T_D) - 1]^{-1}$, where $G_{Th}$ is the total thermal conductance of each suspended device (see Fig. 2a), which encompasses conduction through the support beams and radiative exchange with the gate (see "Methods" for details of how $G_{Th}$ is determined at each gap size).

Next, we repeated this process for a range of gap sizes (from 25 μm to contact) and gate temperatures (25 °C to 117 °C) starting with a initial $\Delta T_S$ value of (~4.9 K) when the gate device was at a gap size of 25 μm. We note that in performing these experiments, the gap distance between the devices was controlled using a combination of a piezoelectric actuator and a stepper motor mounted on a translational

stage. In Fig. 2b, we illustrate the approach process with a time trace where the gap size between the gate and the source-drain devices was systematically reduced in steps of ~ 0.35 μm from ~ 8 μm to contact (a shorter range of gap sizes is shown for clarity). The values of $\Delta T_S$ and $\Delta T_D$, at a $T_G$ of 25 °C, were simultaneously recorded for each gap separation, $d$. The middle and bottom panels of Fig. 2b show the simultaneously measured temperature changes in the source ($\Delta T_S$) and drain ($\Delta T_D$) devices, respectively. The point of physical contact between the gate and the source-drain electrodes was indicated by a large simultaneous change in the temperature of both the source and drain.

To calculate $G_{S-D}$ (see above), we use the measured values of $\Delta T_S$ and $\Delta T_D$ for various gap sizes between the source-drain device and the gate and for various temperatures of the gate electrode. The results for the full range of gap sizes are shown in Fig. 2c. The data shown in dark blue in Fig. 2c correspond to the situation where the gate is at room temperature (25 °C), i.e., when the $VO_x$ film is in the insulating phase. For this case, the measured conductance was ~ 240 pW/K for a large gap separation (~ 25 μm) between the source-drain device and the gate. As the gap distance was reduced to ~ 16.5 μm, we observed a modest increase in the conductance to a maximum value ($G_{S-D,max}$) of 257.2 pW/K. Interestingly, as reported in a recent study[34], the measured conductance is much larger than that expected from the blackbody limit (~ 4 pW/K) calculated from far-field radiative heat theory (FF-RHT), see Supplementary Note 7 for more details of this estimate of the blackbody limit. As the gap size was decreased further to ~ 10 μm, there was a gradual decline in $G_{S-D}$, followed by a more rapid decrease for even smaller gaps. This more rapid decrease in $G_{S-D}$ below ~ 10 μm occurs due to the strong influence and interaction of the gate on the guided modes supported by the source and drain membranes[39], which is discussed in more detail below. Finally, we note that the minimum value of the source-drain thermal conductance ($G_{S-D,min}$), ~ 13 pW/K, was reached just before contact between the gate and the source-drain device.

Similar behavior in the gap dependence of $G_{S-D}$ was found for a range of gate temperatures ($T_G = 33$ °C and 54 °C, respectively), as shown in Fig. 2c. As the gate temperature approached the phase transition temperature (~ 68 °C), a noticeable change in thermal conductance was observed, confirming that the $VO_x$ began to undergo a transition from an insulating state to a metallic state. Once the gate temperature was increased to 83 °C or higher, where $VO_x$ is in its metallic state, the conductance increased to a higher value at every gap size (see Fig. 2c). Specifically, at these higher gate temperatures, it was observed that when $d$ is 25 μm, $G_{S-D}$ is ~ 310 pW/K, which is larger than the conductance observed when the gate was in the insulating phase. This is likely because the gate has a higher reflectivity in the metallic phase, resulting in an enhancement of the number of photons received by the drain electrode[40]. At these high temperatures, as $d$ was further reduced, $G_{S-D}$ decreased continuously, reaching a minimum value of ~ 39 pW/K right before contact between the gate and the source-drain device.

The above-described experiments reveal that the heat current between the source and drain can be controlled by either changing the phase of the $VO_x$ film integrated into the gate or by changing the gap distance $d$ between the gate and the source-drain device. Past work[16,24,41,42] has referred to devices that allow control of heat flow between two objects via a third object (including the introduction of ions as the electrolyte) as a thermal transistor. Following this convention, the device discussed in this work also represents a thermal transistor in which the heat current between the source and the drain is controlled by a third body (gate). In order to make this explicit, in Fig. 3a we symbolically represent our device as a thermal transistor where the gate plays a role similar to the gate electrode of an electrical transistor and controls the flow of heat between the source and the drain, which are analogous to the source and drain electrodes of an electrical transistor.

In order to directly illustrate this switching action, we apply a bipolar voltage (ranging from 0 V to ± 4.0 V) to the resistor integrated into the gate that enables varying the temperature of the gate from 25 °C to 117 °C. At any given gap size, the "Off" state of the thermal transistor corresponds to the situation where the $VO_x$ gate is in the insulating phase, resulting in a lower heat current between the source and the drain, while the "On" state corresponds to the case where the gate is in the metallic phase, which results in a higher heat current between the source and the drain. In Fig. 3b, we show the On/Off ratio ($R = G_{S-D,on}/G_{S-D,off}$) as a function of $d$. We found that for $d = 25$ μm the On/Off ratio was ~ 1.31. As $d$ was reduced, a noticeable increase in the On/Off ratio was observed with the ratio reaching 3.02 right before contact. The noticeable disagreement between the experimental and simulated data (details of simulations are discussed later) in Fig. 3b for gap sizes of around 5 μm likely arises from the simplifications in the model geometry used in our simulation and, to a lesser degree, from the uncertainty in the dielectric functions.

In order to directly demonstrate heat current switching, we simultaneously recorded the radiative conductance while rapidly increasing and decreasing the gate temperature. Specifically, we performed an experiment when the temperature difference between the source and the drain was ~ 4.4 K, with a fixed gap size of $d$ ~ 1 μm or less (just before contact). We chose this minimal gap size to achieve the largest On/Off ratio. Modulation of the gate device temperature was accomplished by applying a bipolar square wave voltage (± 4 V for 117 °C, 0 V for 25 °C) with a period of 200 s (Fig. 3c, bottom panel). The measured conductance between the source and the drain (Fig. 3c, top panel) is consistent with the results from Fig. 2c. The radiative conductance changes by a factor of three, i.e., from ~ 13 pW/K (insulator phase) to ~ 39 pW/K (metallic phase). At this smallest gap, we also performed experiments to systematically explore the effect of gate temperature on the heat current between the source and drain. Results from this experiment are shown in Fig. 3d. As expected, when the gate temperature was below the phase transition temperature (around 68 °C), the heat flow between the source and drain electrodes was low and remained relatively constant even as the gate temperature increased. However, once the phase transition temperature was surpassed, a sudden jump in the heat transfer rate was observed. In addition, we also performed experiments while cooling the gate (Fig. 3d), which revealed hysteretic behavior in the heat flow that is consistent with the hysteresis of the phase transition properties of $VO_x$[43,44]. The hysteresis is primarily caused by structural changes within the material during its phase transition. These changes have been explored via MD simulations, attributing the observed change to significant alterations in its crystal lattice structure and electron concentration[45,46].

The above experiments reveal that for our thermal transistor, the maximum $R$ value is ~ 3, which is comparable to that demonstrated in past work. For example, Cho et al. reported an On/Off ratio of 1.5 with a switching time of 1.5 h during room temperature operation[41]. A. Sood et al. demonstrated an On/Off ratio of approximately 10 with a switching time of 7 min under the same operating temperature[26]. Q. Yang et al. reported an On/Off ratio of 4 with a switching time of 3 min at an operating temperature of 280 °C[24]. L. Castelli et al. demonstrated the fastest switching speed for a thermally gated transistor, with a time constant of 1.7 minutes when the device was turned off, and a switching speed of 4.6 min when the device was turned on, and reported an On/Off ratio of 109[25]. While the $R$ values reported by us are smaller than in some previous works, the switching speed of our devices is orders of magnitude faster as quantified below.

The switching speed of thermal transistors is primarily controlled by the thermal time constant of the source, drain, and gate electrodes of the transistor. To quantify the switching speed for our devices, we performed frequency response measurements, where we measured the amplitude of temperature oscillations as a function of the

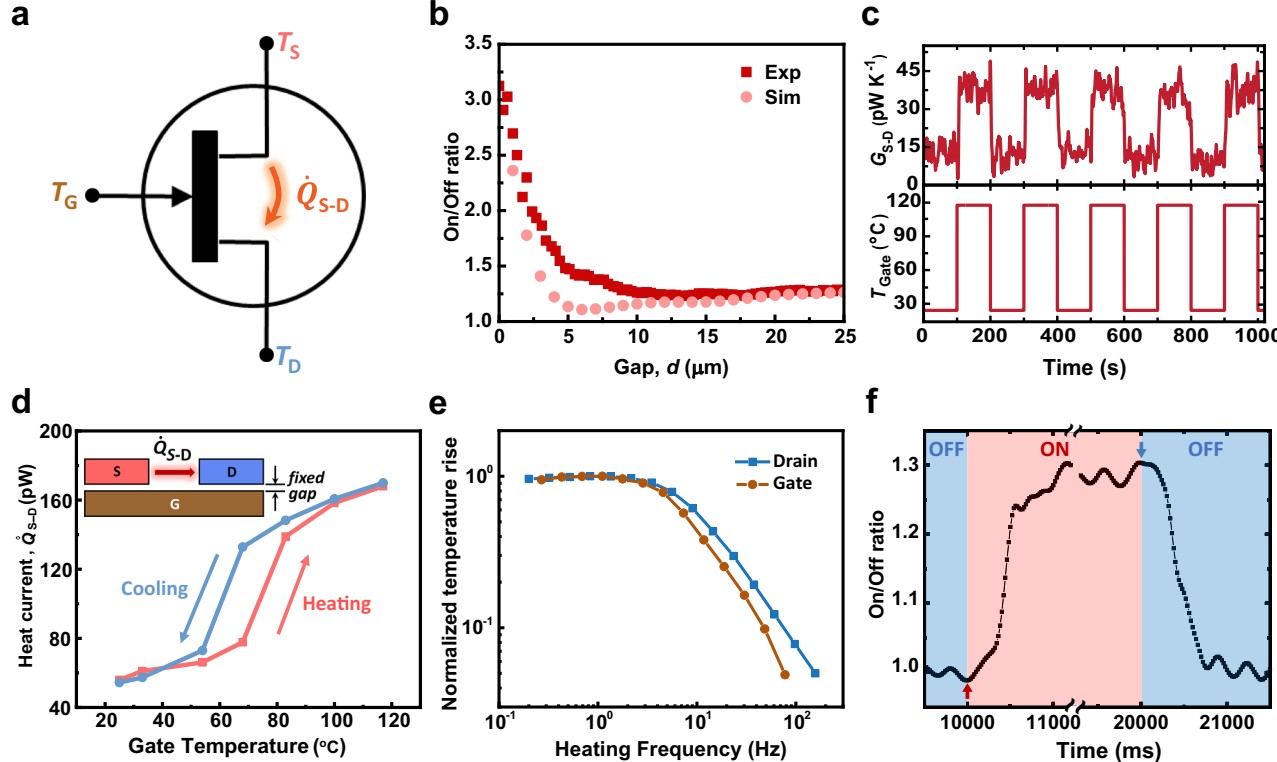

**Fig. 3 | Characterization of thermal transistor performance. a** Symbol representing the designed thermal transistor, illustrating control of the heat current between the source and drain electrodes via the gate temperature. $T_S$, $T_D$, and $T_G$ denote the temperatures of the source, drain, and gate, respectively, while $\dot{Q}_{S-D}$ indicates the heat current between the source and the drain. **b** On/Off ratio of the thermal transistor as a function of the gap size, $d$. **c** Measured change in the source-drain conductance ($G_{S-D}$, top panel) when the temperature of the gate ($T_{Gate}$, bottom panel) periodically changes from 25 °C to 117 °C and back by applying a bipolar voltage from 0 V to ± 4 V. In this experiment, the period of the gate temperature variation is 200 s, and the gate is positioned just before the contact ($d$ of less than ~ 1 μm). **d** Heat current ($\dot{Q}_{S-D}$) between the source and the drain as a function of the gate temperature just before contact ($d <$ ~ 1 μm). The temperature difference between the source and the drain is measured to be ~ 4.4 K before contact. **e** Measured thermal frequency response of the drain and gate devices. **f** Measured time response with a lock-in time constant of 100 ms. These data show a switching time ($\tau_{Off-On}$) of ~ 470 ms for the rise and $\tau_{On-Off}$ of ~ 500 ms for the decay response, respectively.

**Table 1 | Summary of some of the thermal transistor and thermal switch characteristics from past and current works**

| Study | Gating mechanism | Operating temperature | Switching ratio | Switching time |
|---|---|---|---|---|
| J. Cho et al., ref. 41 | Electrochemical | Room Temperature | ~1.5 | ~1.5 h |
| A. Sood et al., ref. 26 | Electrochemical | Room Temperature | ~10 | ~ 7 min |
| Q. Yang et al., ref. 24 | Electrochemical | 280 °C | ~ 4 | ~ 3 min |
| M. Li et al., ref. 47 | Electric Field | Room Temperature | ~13 | ~1 μs |
| L. Castelli et al., ref. 25 | Thermal | Room Temperature | 109 ± 44 | $\tau_{OFF-ON}$ = 1.7 min $\tau_{ON-OFF}$ = 4.3 min |
| Y. Li et al., ref. 27 | Thermal | Room Temperature | 14.3 | — |
| Current work | Thermal | Room Temperature | ~ 3 | $\tau_{OFF-ON}$ = ~ 470 ms $\tau_{ON-OFF}$ = ~ 500 ms |

frequency of heat input (see "Methods" for details). The measured frequency responses of the drain (nominally equivalent to the source) and the gate device are shown in Fig. 3e. The drain device features a cutoff frequency ($f_c$) of 6.89 Hz, and the gate device has a cutoff frequency of 5.33 Hz, respectively. Therefore, the thermal time constant ($\tau = 1/2\pi f_c$) of the drain is ~ 23 ms, while that of the gate is ~ 30 ms. This analysis suggests that the switching time of the photonic thermal transistor is expected to be at least ~ 30 ms, which is limited by the slower thermal response of the gate device. To experimentally determine the switching speed of our thermal transistor device, we performed an experiment (see "Methods") with a significantly shorter lock-in time constant (100 ms). Data corresponding to this experiment is shown in Fig. 3f and features a switching speed of ~ 500 ms, which is a much faster response time than that demonstrated to date in thermal transistors. However, this

switching speed is still slower than that expected from the frequency response measurements described above and is limited by the lock-in time constant. Reducing the lock-in time constant further (less than 100 ms) was not feasible because of the limited signal-to-noise ratio of the measurements and capacitive voltage spikes generated by the square wave modulation technique. Details of the transient time response measurement procedure are explained in the Method section. Finally, we note that as opposed to the three-terminal thermally-gated transistors presented here, recent work by M. Li et al. has explored electric field-controlled two-terminal molecular self-assembled monolayers (SAMs) as thermal switches. Their results suggest that the interfacial thermal resistance of the molecule-electrode junction (measured using thermoreflectance) features an On/Off ratio of 13 at switching speeds of 1 MHz[47]. To compare the performance of our thermal transistor with the other thermal

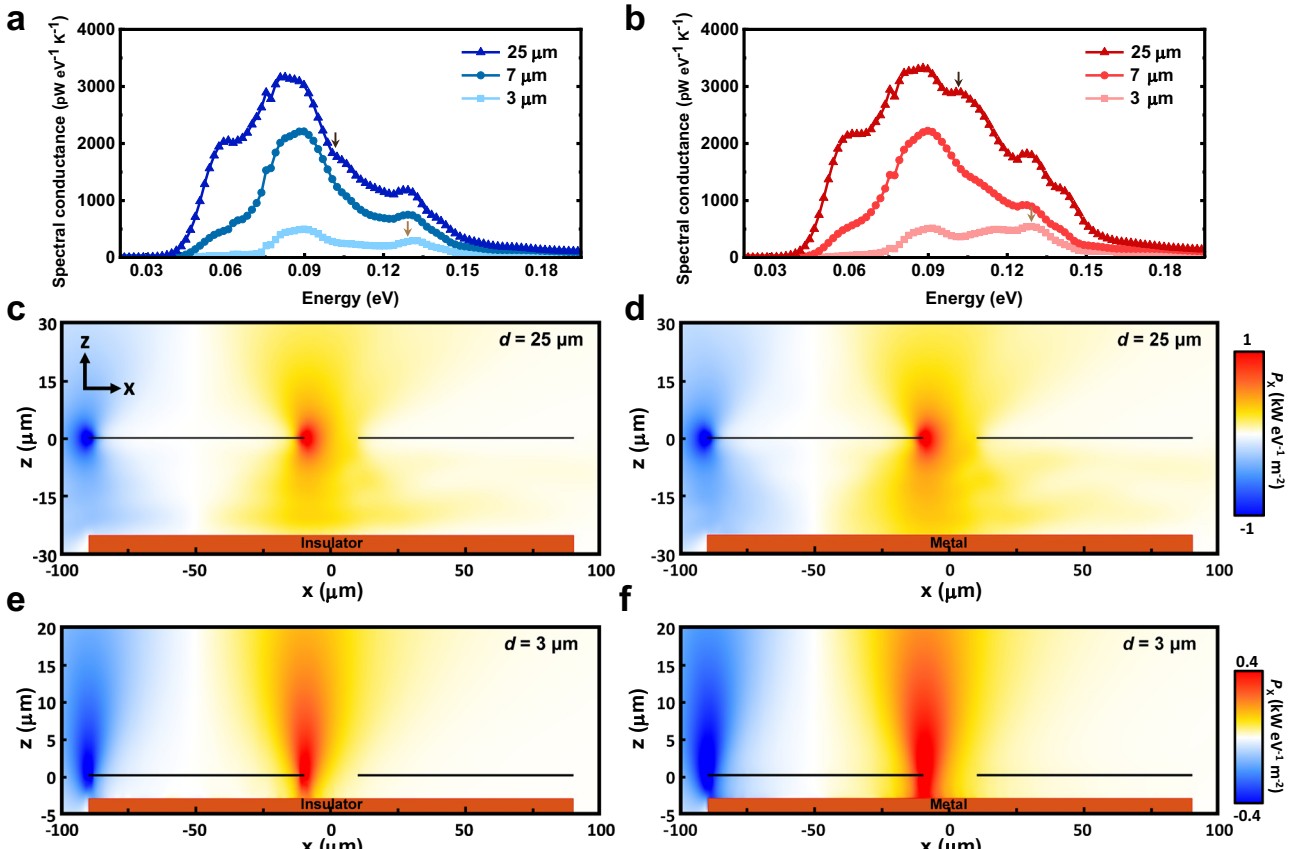

**Fig. 4 | Analysis of spectral conductance and Poynting vector in the three-body system.** The computed spectral conductance plots for the insulating (**a**) and metallic (**b**) phases are shown for a gate at various gap sizes: 25 μm, 7 μm, and 3 μm, respectively. The attenuating effect of the third body becomes clearly visible as it is brought closer to the membranes. The arrows in panels (**a**) and (**b**) indicate the energies at which the Poynting vector plots are calculated in Fig. 4c−f. **c** The Poynting vector in the *x*-direction ($P_x$) at midplane (a plane cutting through the middle of the system), corresponding to an energy of 0.1 eV, and a gap separation of 25 μm between the gate and the top device (with the gate in the insulating phase). **d** Same as (**c**) but with the gate in the conducting phase. **e** Same as (**c**) but the Poynting vector at a frequency of 0.1286 eV and a gap separation of 3 μm. **f** Same as (**e**) but with the gate in the conducting phase. In Fig. 4c−f, the black lines indicate the 250 nm SiN layers (left: Source, right: Drain), while the brown block represents the gate coated with 150 nm VO$_x$.

transistors and switches, we summarize results from multiple works in Table 1.

In order to mechanistically understand the thermal switching phenomena due to the phase transition of VO$_x$, we performed fluctuational electrodynamics-based calculations using SCUFF-EM. We modeled the source-drain device as two coplanar SiN membranes, each measuring 80 μm in length, 60 μm in width, and 250 nm in thickness, with a separation of 20 μm (see Supplementary Fig. 4b). The dimensions of the source-drain device used in the theoretical calculation exactly match those employed in the experiment. The gate was modeled as a 10 μm thick, doped silicon membrane encapsulated with a 150 nm-thick VO$_x$ layer. The dimensions of the gate were chosen to be 180 μm length and 60 μm width (see methods for a discussion of the choice of these dimensions). In our calculations, the gap size (*d*) between the source-drain device and the gate is varied from 25 μm to close to contact (a gap size of 1 μm). The radiative conductance is calculated as a function of gap sizes, both when the VO$_x$ is in the insulator and in the metallic phase. The computed results (Fig. 2d) exhibit a very similar gap size dependence to what was observed experimentally and reveal a lower thermal conductance for both phases, although the absolute magnitude is consistently lower by ~30%. We attribute this discrepancy to the radiative conductance between the support beams and the mismatch of the dielectric function of materials between models and microfabricated devices, which was also observed in our previous works[34,40,48,49].

To gain insight into the physical mechanism underpinning the change in heat flux due to the phase transition of VO$_x$, we calculated the spectral conductance at various gap sizes for both the insulating and metallic phases (Fig. 4a, b). It can be observed that, as the gap size is reduced, there is a strong attenuation in the spectral conductance at all frequencies for both insulating and metallic phases, resulting in a reduction of heat transfer in both cases. However, the insulating phase does exhibit a more rapid attenuation as the gap size decreases when compared to the metallic phase.

In order to understand the effect of the gate on $G_{S-D}$, we computed the in-plane Poynting fluxes in the *x*-direction ($P_x$), which is the direction of heat flow from the source to the drain. The Poynting fluxes were calculated at frequencies where the difference in the value of spectral conductance between the two phases is largest (i.e., 0.1 eV when *d* = 25 μm; 0.1286 eV when *d* = 3 μm), thus making the contrast between the heat flow of different phases more apparent (see arrows in Fig. 4a, b). When the gate is far away (e.g., *d* = 25 μm), the Poynting flux appears very similar for both the insulating and metallic phases, with a slight enhancement in the heat flux of the metallic phase due to the higher reflection on the metallic surface (Fig. 4c, d). However, when *d* = 3 μm, we observe a noticeable enhancement in the Poynting flux for the case where the gate is metallic, in comparison to the insulating phase (Fig. 4e, f). This difference in radiative conductance at small gap sizes arises from the fact that, for a metallic layer, there is very poor radiative coupling between the top device and the gate. This poor

radiative coupling, in turn, results in a smaller attenuation in heat transfer between the source and drain membranes[39].

To summarize, we present a nanoscale radiative thermal transistor capable of controlling the heat current between two nanometer-thick membranes. This transistor enables precise manipulation of heat flow via a gate, whose dielectric properties can be tuned by modifying the temperature. Specifically, we demonstrate that the radiative heat transfer between the source and drain membranes can be modified by up to a factor of three when the gate undergoes a metal-insulator transition. More interestingly, our nanomembrane-based thermal transistor exhibits fast switching times (~500 ms), attributed to the small thermal mass of our devices, which is two orders of magnitude faster than the thermal transistors presented in a recent work[25]. Our experimental findings are complemented by a theory-based model calculation that elucidates the mechanism of thermal modulation. Taken together, these results highlight the potential for fast heat current modulation via nanoscale membranes. These findings have the potential to make a substantial impact on the fields of thermal management and computing, particularly in the areas of heat current control, thermal circuits and logic devices, and thermal energy conversion.

## Methods

### Measurement of the source and the drain temperature

The source-drain device is oriented in parallel with respect to the gate device in a high vacuum chamber (pressure $< 10^{-6}$ Torr) using a custom-built nanopositioner[38]. During the measurements, the drain is nominally at room temperature ($T_{Amb}$ ~ 298 K), while the source temperature is modulated by supplying a fixed amplitude $I_{AC} = 12$ μA sinusoidal current at a frequency of 1 Hz to the platinum heater line integrated into the suspended source membrane. The applied current isothermally heats up the source by ~ 4.9 K at a frequency of 2 Hz. The frequency of 1 Hz was selected as this frequency was sufficiently low to achieve a full thermal response, as seen in Fig. 3e, where the normalized temperature response is shown as a function of frequency. The amplitude of the temperature oscillation of the source ($\Delta T_S$) at 2 Hz was estimated by monitoring the voltage oscillation at 3 Hz ($\Delta V_S$) across the Pt heater line with an SR830 lock-in amplifier. The temperature fluctuation of the source ($\Delta T_S$) at 2 Hz was determined from the measured voltage ($\Delta V_S$) following the equation: $\Delta T_S = 2\Delta V_S / I_{AC} R\alpha$, where $R$ is the electrical resistance of the Pt heater (~ 16.8 kΩ), and $\alpha$ is the TCR of the Pt heater ($1.75 \times 10^{-3}$ K$^{-1}$). More details of the electrical response, TCR measurement, and beam conductance measurement of the source are provided in Supplementary Note 5.

The heat flux from the source to the drain results in temperature oscillations ($\Delta T_D$) of the drain at 2 Hz. In order to measure ($\Delta T_D$), a direct current $I_{DC} = 10$ μA was applied across the Pt thermometer that is integrated into the drain device. Voltage oscillations ($\Delta V_D$) at a frequency of 2 Hz were measured with a SR830 lock-in amplifier. Finally, $\Delta T_D$ was determined from the equation $\Delta T_D = \Delta V_D / I_{DC} R\alpha$, where $R$ and $\alpha$ are the electrical resistance and the TCR, respectively, for the drain Pt thermometer.

### Measurement of gate temperature

To determine the temperature of the gate, we first measured the temperature coefficient of resistance (TCR) of the gate device under vacuum conditions in a cryostat. The TCR was determined by observing the change in resistance under a current ($I_{AC} = 10$ μA, $f = 101$ Hz) to ensure negligible self-heating while varying the temperature using a cryostat (Janis ST-100). We then proceeded to measure the resistance of the gate device while applying a bipolar voltage ranging from 0 V to $\pm 4.0$ V. By combining these measurements, we were able to estimate the temperature of the gate device at different power values, which in turn provided us with the beam conductance at each temperature. The temperature of the gate was measured at various bipolar voltages:

25 °C at 0 V, 33 °C at $\pm 1.0$ V, 54 °C at $\pm 2.0$ V, 68 °C at $\pm 2.5$ V, 83 °C at $\pm 3.0$ V, 100 °C at $\pm 3.5$ V, and 117 °C at $\pm 4.0$ V, respectively, with a temperature uncertainty $< \pm 1$ K for each measurement. A more detailed method for measuring the gate temperature is explained in Supplementary Note 6.

### Evaluation of $G_{Th}$ and expression for radiative conductance ($G_{S-D}$)

To evaluate the radiative thermal conductance ($G_{S-D}$) between the source and the drain, a lumped thermal model was used. In this model, $G_{Th}$ represents the thermal conductance of the source as well as the drain device and is dominated by the conduction through the beams but has some contributions from radiative exchange with the gate and the ambient. The characteristics of the source and drain devices are nearly identical since they were fabricated using the same procedure; hence, $G_{Th}$ is almost equal for both devices (Supplementary Note 5). It should be noted that $G_{Th}$ mostly comprises of thermal conduction through the beams ($G_{beams}$ ~ 250 nW/K as shown in Supplementary Fig. 5c), with a minor influence from near-field coupling between the source and gate. As the gap gets smaller, the near-field contribution can reach magnitudes of about 10 – 30 nW/K, depending on the gate temperature, which means that the radiative exchange between each membrane and the gate depends on the gap size. Thus, the value of $G_{Th}$ for each gap $d$ can be calculated using the equation $G_{Th} = P_{Joule}/\Delta T_S$, where $P_{Joule}$ represents the power dissipated in the Pt heater ($P_{Joule} = I_{AC}^2 R/2$), $R$ is the resistance of serpentine platinum, $\Delta T_S$ indicates the temperature change of the source ($\Delta T_S = 2\Delta V_S / I_{AC} R\alpha$), and $I_{AC}$ is an alternating current (we chose $I_{AC} = 12$ μA at $f = 1$ Hz). Based on the lumped thermal model illustrated in Fig. 2a, the radiative heat flow to the drain can be defined as $Q_{S-D} = G_{Th} \times \Delta T_D$. Accordingly, the radiative conductance between the source and drain device is given by $G_{S-D} = Q_{S-D}/(\Delta T_S - \Delta T_D)$, or equivalently $G_{S-D} = (G_{Th}) \times [(\Delta T_S/\Delta T_D) - 1]^{-1}$.

### Thermal frequency response of the source and gate devices

To analyze the thermal frequency response of the source devices, a sinusoidal current with a fixed amplitude of $I_f = 6$ μA was passed through the integrated platinum resistance thermometer (PRT) to induce Joule heating. The frequency of the supplied current was varied within a wide range, from 0.1 Hz to 100 Hz, causing corresponding heat $Q_{2f} = I_f^2 R/2$ across the integrated Pt heater at $2f$. This heat dissipation led to a temperature fluctuation ($\Delta T_{2f}$) in the source, which was determined by measuring the $3f$ component of the voltage ($\Delta V_{3f}$) across the PRT using a lock-in amplifier. The temperature fluctuation can be calculated using the following equation:

$$\Delta T_{2f} = \frac{2\Delta V_{3f}}{I_f R\alpha} \tag{1}$$

where $R$ and $\alpha$ denote the resistance and TCR of the device, respectively. Figure 3e illustrates the normalized temperature rise of the source device. A complete thermal response is achieved at $2f$ below approximately 2.6 Hz, beyond which the signal begins to roll-off. The measured cutoff frequency of the source device was found to be 6.89 Hz.

To measure the thermal frequency response of the gate device, we employed a similar approach to that described above, except for the amplitude of the current (see Fig. 3e). Specifically, a sinusoidal current with a fixed amplitude of $I_f = 1$ mA was applied through the Si serpentine, resulting in Joule heating of the gate device at a frequency of $2f$. A larger current is used in this experiment as the beam thermal conductance of the gate (~ 0.4 mW/K at room temperature) is much larger compared to the source device (~ 250 nW/K), thus more current is required to achieve a comparable temperature raise in the gate device. The frequency of the supplied current was varied across a wide range, from 0.1 Hz to 100 Hz. For the gate, a full thermal response was achieved at frequencies below ~ 2 Hz, beyond which the signal begins

to roll-off. The measured cut-off frequency of the gate device was 5.33 Hz.

## Transient time response measurement

To experimentally determine the thermal response time of the device, we modulated the gate temperature using a square-wave approach. The temperature was varied from 330 K to 354 K with a period of 20 seconds and a 50% duty cycle. The square wave rise and fall time was rolled off to ~ 300 ms to reduce any capacitive voltage spikes in the measurement. The source was heated using an AC current with an amplitude of 12 μA and a heating frequency of 6 Hz to enable a fast thermal response in the experimental setup. We collected the temperature data of the source and drain from this measurement at a sampling rate of 50 Hz for a total duration of 12 h using an integration time of 100 ms on the lock-in amplifier (SR830). Throughout the measurement process, the signal exhibited consistent stability across >1000 switching cycles, showing no discernible variation over thousands of iterations. Finally, to improve the signal-to-noise ratio, we averaged the temperature data over ~ 2160 periods. Using a lumped capacitance analysis, we estimated the thermal time constant (switching time) to be the interval when the radiative conductance has changed by $1 - e^{-1}$ (~ 63.2 %) of its total change. We see from our measurements that we achieved an off-on switching time ($\tau_{\text{Off-On}}$) of ~ 470 ms and an on-off switching time ($\tau_{\text{On-Off}}$) of ~ 500 ms. This shows that this thermal transistor does indeed have a very symmetric, sub-second switching characteristic.

## Computing the radiative conductance and Poynting fluxes in the many-body system with SCUFF-EM

The source and drain membranes were modeled as 80 μm long, 60 μm wide, and 250 nm thick silicon nitride (SiN) membranes. The gate was modeled as a 180 μm long, 60 μm wide, and 10 μm thick doped silicon substrate with 150 nm $VO_x$ coated on all sides. These dimensions were chosen to keep the computations tractable without altering the physics of the problem. We chose to model the gate with $VO_x$ coating on all sides as we were not able to get a converged solution with $VO_x$ only present on the top layer, which was most likely caused by the computational challenges of three-material junctions (vacuum, doped Si, and $VO_x$). As shown in Supplementary Note 4, the presence of $VO_x$ on all the other sides except the top surface plays little to no role in the heat transfer from the source to the drain, mainly because the 10 μm-thick doped Si layer becomes opaque to all thermal radiation.

For the Poynting flux calculations, we changed the width of all the membranes, i.e., the source, drain, and gate, from 60 μm to 30 μm. These calculations are computationally demanding and thus a full-scale geometry could not be modeled. Thermally induced surface current sources were setup in the source at 300 K and the spatial distribution of the Poynting flux was evaluated at 0.1 eV and 0.1286 eV for gap sizes of 25 μm and 3 μm, respectively.

## Data availability

Source data for Figs. 1 to 4 are provided with this paper. Source data are provided with this paper.

## Code availability

The Poynting flux results were calculated using the open-source code SCUFF-EM. COMSOL Multiphysics was used for modeling the conductance of the blackbody limit.

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

## Acknowledgements
We acknowledge support from DOE-BES through a grant from the Scanning Probe Microscopy Division under award No. DESC0004871 (experiments and analysis) and support from the Army Research Office under award No. W911NF-19-1-0279 (fabrication of devices).

## Author contributions
E.M. and P.R. conceived the work. J.W.L. designed the experiments, fabricated the devices, conducted the experiments, and performed COMSOL simulations. A.M. performed the SCUFF-EM calculations and assisted in the experiments. R.M. fabricated the gate device and assisted in the experiments. A.-R. G. and Y.L. contributed to the characterization of material properties. All the authors contributed to the data analysis. The manuscript was written by J.W.L., A.M., E.M., and P.R. with comments and input from all authors.

## Competing interests
The authors declare no competing interests.
