## [Peer Review File · Nature Communications]

A Nanoscale Photonic Thermal Transistor for Sub-Second Heat Flow SwitchingREVIEWER COMMENTS

Reviewer #1 (Remarks to the Author):

In this paper, a nanoscale radiative thermal transistor—a three terminal system consisting of source, drain, and gate devices are experimentally demonstrated. The authors give a physical picture of this thermal transistor and perform detailed tests, which seem to be quite feasible. The related results are also very interesting, I think it's enough to get the attention of researchers in related engineering fields and has potential applications. This manuscript provides sufficient test results and analysis, is well written, and the conclusions are sound. However, before it is accepted for publication, the paper needs to be further improved. Below are the specific comments.

- 1) How did the gate material selection process work, i.e., why was VOX ultimately chosen and can authors give a more detailed explanation?
- 2) Does the frequent phase transition of the VOx affect the gate structure?
- 3) How the environment affects the performance of the thermal transistor, whether this performance is maintained when the external environment, such as temperature, changes too much.
- 4) The authors may consider adding more microscopic simulations such as molecular dynamics and performing mechanistic analyses.

Reviewer #2 (Remarks to the Author):

The authors' research represents a critical step in realizing a near-field thermal transistor with enhanced switching capabilities. Experimentally, they demonstrate the fine control of heat flow between the source and drain by manipulating the phase of the material integrated into a gate or adjusting the distance between the source and drain, similar to a transistor. This demonstration holds significant promise for advancing thermal management and logic devices, offering avenues for more efficient thermal circuits and addressing pressing challenges in the field.

While the pursuit of effective heat management through thermal transistors remains an active area of research, the insights provided by this study mark a notable contribution to understanding thermal modulation mechanisms. Further inquiries into the experimental setup, theoretical foundations, and practical implications of these findings are warranted, shedding light on nanoscale thermal transistors' scalability and real-world applications, so I would like to pose the following questions to the authors.

1. The observed achievement of the transistor operating in near-field and demonstrating faster switching is commendable. How does this transistor compare and integrate to thermal transistors that use solid-state technologies and still use electric fields /magnetic fields(experimental)[1]/optical fields(theory)[2]? Could you comment with respect to the references below:

[1] Li, Man, et al. "Electrically gated molecular thermal switch." *Science* 382.6670 (2023): 585-589

[2] R. T. Wijesekara et al., "Optically controlled quantum thermal gate," *Phys. Rev. B* 101, 245402(2020)

Providing a comprehensive analysis will help contextualize the significance of radiative transistors and their potential impact within the broader landscape of using these in future electronic circuitry or quantum technology applications.

2. Researchers have reported numerous feasible designs, so it is crucial to establish a means for comparing various technologies. Given our interest in comprehending how the device's performance is quantified in terms of efficiency, could the authors propose a figure of merit that facilitates comparison among different designs of radiative thermal transistors? For instance, the reference [1] cited above suggests the term 'tunability to conductance' as a potential metric.

3. Does this study investigate the variances in operational mechanisms among thermal transistors utilizing the same phase transition materials as VOx at the gate? What accounts for the omission of the parameter 'negative differential resistance,' as discussed in Ref.23, which leads to the amplification of the heat flux received by the drain in the proposed transistor design?

[3] Li, Yuxuan, et al. "Radiative Thermal Transistor." *Physical Review Applied* 20.2 (2023): 024061.

4. What is the typical operating temperature range (particularly in relation to the gate temperature) for this device demonstrating transistor characteristics with an optimal conductance (GS-D) with a suitable/optimal gap distance selected (that can be easily fabricated)? I feel the inclusion of this data in the abstract will help readers get a summary of the predicted device characteristics that are provided in Figure 2. Additionally, when experimenting with heat current switching, what is the reason for selection for a particular set of temperature and a gap size?

5. The authors allude to the hysteretic behaviour associated with the phase transition properties of VOx. Such behaviour can pose challenges if the device is intended for switching applications, potentially leading to functionality issues. Could the authors offer insights into any dead band limits associated with changes in source-drain conductance during the cycling of the transistor between its ON and OFF states in the experiment? Moreover, what implications does this behaviour have on the ON/OFF ratio?

Furthermore, it would be beneficial to understand how the authors accounted for noise threshold values, such as signal-to-noise ratio, in characterizing the switching characteristics. Given the inherent complexities of hysteretic behaviour, elucidating its impact on the device's performance and reliability is crucial for assessing its practical applicability.

6. Could the authors expand Figure 3 (d) or provide a comment on the thermal sensitivity of the device with respect to the variation of the heat current rates for the small variation in the temperatures?

7. The authors have not adequately portrayed the current state of thermal transistor technology in their work. The theoretical advancements in this field have progressed rapidly,

encompassing steady-state model [4], periodic drive analyses [5], and stochastic small signal model [6]. It would be beneficial to include a discussion of these advances in the paper, as they provide insights into the physical possibilities for implementation.

[4] Bao-qing Guo et al, Multifunctional quantum thermal device utilizing three qubits, PhysRevE.99.032112.

[5] Nikhil Gupt et al. "Floquet quantum thermal transistor," Phys. Rev. E 106, 024110 (2022)

[6] Uthpala N. Ekanayake et al., "Stochastic model of noise for a quantum thermal transistor," Phys. Rev. B 108, (2023)

Reviewer #3 (Remarks to the Author):

The authors present a compelling study on a nanoscale radiative thermal transistor utilizing a gate structure. They demonstrate control over radiative heat transfer between source and drain via insulator-metal phase transition in the material, significantly impacting guided modes. An on/off ratio of ~ 3 is achieved at a gap size of $\sim 1 \mu\text{m}$ with sub-second switching time.

The experimental setup exhibits high reliability with meticulous system and measurement calibration. The manuscript is clear, well-written, and supported by sound experimental data and simulations. Given the current interest in thermal transistors within the heat transfer community, this timely demonstration will undoubtedly generate significant attention. For the reasons stated above, I recommend this manuscript for publication after minor revision.

Suggestions for improvement:

* Total Power Calculation (Fig. 4): Supplement the spectral conductance data with calculations of total power as a function of gap distance. Compare this data with the measurement results in Fig. 2d. The simulation should also include a comparison of the total power ratio between insulating and metallic phases, along with the gap size where conductance drops (observed at $10 \mu\text{m}$ in the experiment).

* Comparison Table (Page 9): For clarity, create a table summarizing on/off ratios and switching times for easier comparison with previous works mentioned on page 9.

* Discrepancy Explanation (Fig. 3b): Address the noticeable discrepancy between experimental data and simulations around the $5 \mu\text{m}$ gap size in Fig. 3b. Provide an explanation for this difference.

* Oscillation Source (Fig. 3f): Explain the source of the periodic oscillation observed in Fig. 3f.

* Missing Axis Labels (Figs. 4c-f): Include missing axis labels in Figs. 4c-f.

* Terminology: It would be better not to use the term "novel" in the abstract and summary.

RESPONSE LETTER

We thank the reviewers for taking the time to review our manuscript and for providing their valuable feedback. We are also grateful for the insightful comments that helped us to improve the manuscript. We have now thoroughly revised the paper, addressed all the issues raised by the reviewers, and made substantial changes. We feel these changes have significantly improved the quality of the paper. Below, we provide a detailed point-by-point response to each of the comments of the reviewers and point out the changes that we have made to the manuscript. All the changes made to the manuscript that are copied in the response letter are marked in yellow in quotations.

Report of Referee A

In this paper, a nanoscale radiative thermal transistor—a three terminal system consisting of source, drain, and gate devices are experimentally demonstrated. The authors give a physical picture of this thermal transistor and perform detailed tests, which seem to be quite feasible. The related results are also very interesting, I think it's enough to get the attention of researchers in related engineering fields and has potential applications. This manuscript provides sufficient test results and analysis, is well written, and the conclusions are sound. However, before it is accepted for publication, the paper needs to be further improved. Below are the specific comments.

We thank the referee for taking the time to review our manuscript. We appreciate the referee's positive feedback regarding the significance and potential impact of our work in demonstrating a nanoscale radiative thermal transistor. Below, we address all the questions raised by the referee.

(1) How did the gate material selection process work, i.e., why was VO_x ultimately chosen and can authors give a more detailed explanation?

We chose vanadium oxide (VO_x) as a gate material for three reasons:

- 1) VO_x exhibits rapid and reversible phase transitions. For example, VO_x can undergo phase transitions at the nanosecond to sub-nanosecond time scale^{1,2}. Such short phase transition times permit operation where the intrinsic thermal time constant of the devices poses the primary limit to the switching times of the transistor, enabling operation at the ultimate switching speed achievable for a given source and drain.
- 2) VO_x features a reversible phase transition at a relatively low temperature (~340 K) compared to other phase-transition materials (e.g., Ti₃O₅, LaCoO₃, Ti₂O₃, NbO₂), whose transition temperatures range from 448 to 1,081 K³. This feature also makes the choice of VO_x attractive.
- 3) Relative ease of integration of high-quality VO_x thin film onto micro-calorimeters via microfabrication.

(2) Does the frequent phase transition of the VO_x affect the gate structure?

We thank the referee for this question. We point out that we have performed experiments where we switched the phase of VO_x >1000 times with no observable change in the switching properties. This points to the robustness of the system. To highlight this point, we now state on page 17 of the manuscript:

“Throughout the measurement process, the signal exhibited consistent stability across >1000 switching cycles, showing no discernible variation over thousands of iterations.”

(3) How the environment affects the performance of the thermal transistor, whether this performance is maintained when the external environment, such as temperature, changes too much.

We note that all our experiments were performed at room temperature and under a vacuum ($\sim 10^{-6}$ torr). We do not foresee any challenges in operation if the room temperature drifted by a few Kelvins. However, if the ambient temperature is much larger than room temperature then a different phase change material, which has phase transition temperatures at a suitable temperature may have to be chosen to implement the thermal transistor.

(4) The authors may consider adding more microscopic simulations such as molecular dynamics and performing mechanistic analyses.

We thank the referee for providing this suggestion. We believe that microscopic simulations, such as new molecular dynamics (MD) simulations, while potentially beneficial do not necessarily add much to the interpretation or understanding of the operation of our thermal transistor. This is particularly true as others^{4,5} have already explored the physics of the VO_x phase transition mechanisms via MD simulations and repeating such calculations is unlikely to add new insights. For example, D. Wang *et al.*⁴ utilized MD simulations to confirm the presence of phase transformation of VO_x, demonstrating that vanadium oxide transitioned completely from a semiconductor phase to the metallic phase in the temperature range of 278 K to 353 K. Furthermore, Y. Ma *et al.*⁵ demonstrated the construction of a full-scale computational model using molecular dynamics to analyze the phase transition mechanism of VO_x nanofilms. Their experimental observations and numerical simulations were in good agreement, confirming the feasibility and accuracy of our full-scale nano-thermodynamics model.

To address this comment, we now state on page 9 of the manuscript:

“The hysteresis is primarily caused by structural changes within the material during its phase transition. These changes have been explored via MD simulations, attributing the observed change to significant alterations in its crystal lattice structure and electron concentration^{45,46}.”

Report of Referee B

The authors' research represents a critical step in realizing a near-field thermal transistor with enhanced switching capabilities. Experimentally, they demonstrate the fine control of heat flow between the source and drain by manipulating the phase of the material integrated into a gate or adjusting the distance between the source and drain, similar to a transistor. This demonstration holds significant promise for advancing thermal management and logic devices, offering avenues for more efficient thermal circuits and addressing pressing challenges in the field.

While the pursuit of effective heat management through thermal transistors remains an active area of research, the insights provided by this study mark a notable contribution to understanding thermal modulation mechanisms. Further inquiries into the experimental setup, theoretical foundations, and practical implications of these findings are warranted, shedding light on nanoscale thermal transistors' scalability and real-world applications, so I would like to pose the following questions to the authors.

We thank the referee for taking the time to provide detailed feedback and for acknowledging the promise of our work. Below we address all the questions raised by the referee.

(1) The observed achievement of the transistor operating in near-field and demonstrating faster switching is commendable. How does this transistor compare and integrate to thermal transistors that use solid-state technologies and still use electric fields /magnetic fields(experimental)[1]/optical fields(theory)[2]? Could you comment with respect to the references below:

[1] Li, Man, et al. "Electrically gated molecular thermal switch." Science 382.6670 (2023): 585-589

[2] R. T. Wijesekara et al., “Optically controlled quantum thermal gate,” *Phys. Rev. B* 101, 245402(2020)

Providing a comprehensive analysis will help contextualize the significance of radiative transistors and their potential impact within the broader landscape of using these in future electronic circuitry or quantum technology applications.

We thank the referee for this question. The comparison to these devices is challenging, as the work by M. Li *et al.* controls interfacial thermal conductance, and in its physical implementation, it represents a two-terminal device. The work by R. T. Wijesekara et al., while very interesting, is theoretical, and it is unclear to us what the physical realization of such a device would be and what the switching ratios would be. However, to acknowledge these contributions we now cite the first reference on page 10 of our manuscript as:

“...recent work by M. Li *et al.* has explored electric field controlled two-terminal molecular SAMs as thermal switches. Their results suggest that the interfacial thermal resistance of the molecule-electrode junction (measured using thermoreflectance) features an On/Off ratio of 13 at switching speeds of 1 MHz⁶.”

We also added the second reference in a different context on page 2 of the manuscript (as explained later in this response letter) where we now state:

“More recently, several theoretical works have also explored the implementation of thermal transistors via quantum effects²⁰⁻²³”

(2) Researchers have reported numerous feasible designs, so it is crucial to establish a means for comparing various technologies. Given our interest in comprehending how the device's performance is quantified in terms of efficiency, could the authors propose a figure of merit that facilitates comparison among different designs of radiative thermal transistors? For instance, the reference [1] cited above suggests the term 'tunability to conductance' as a potential metric.

We thank the referee for this comment. The on/off ratio and switching times are suitable metrics for comparison. Therefore, we have now added a table to our manuscript where the on/off ratios and switching speed are reported (page 24).

Table 1. Summary of some of the thermal transistor and thermal switch characteristics from past and current works.

Study	Gating mechanism	Operating temperature	Switching ratio	Switching time
J. Cho et al. , Ref. [41]	Electrochemical	Room Temperature	~1.5	~1.5 h
A. Sood et al. , Ref. [26]	Electrochemical	Room Temperature	~10	~7 min
Q. Yang et al. , Ref. [24]	Electrochemical	280 °C	~4	~ 3 min
M. Li et al. , Ref. [47]	Electric Field	Room Temperature	~13	~1 μs
L. Castelli et al. , Ref. [25]	Thermal	Room Temperature	109 ± 44	$\tau_{\text{OFF-ON}} = 1.7 \text{ min}$ $\tau_{\text{ON-OFF}} = 4.3 \text{ min}$
Y. Li et al. , Ref. [27]	Thermal	Room Temperature	14.3	—

(3) Does this study investigate the variances in operational mechanisms among thermal transistors utilizing the same phase transition materials as VO_x at the gate? What accounts for the omission of the parameter 'negative differential resistance,' as discussed in Ref.23, which leads to the amplification of the heat flux received by the drain in the proposed transistor design?

[3] Li, Yuxuan, et al. "Radiative Thermal Transistor." Physical Review Applied 20.2 (2023): 024061.

We thank the referee for the question. While our current work and Ref. 23 both employ VO_x as a gate, the physical implementation of the device is very different in these two works. Specifically, the heat flow between the source and drain occurs through the gate electrode in Ref. 23, whereas this is not the case in our work, which employs near-field effects. Therefore, the negative differential resistance observed in Ref. 23 does not arise in our work. We do not venture into discussing this difference, as this would require a detailed discussion of the work of Ref. 23, which would be distracting and potentially confusing to future readers. We will, however, discuss these differences in a future review article where we will summarize all work in this area.

(4) What is the typical operating temperature range (particularly in relation to the gate temperature) for this device demonstrating transistor characteristics with an optimal conductance (GS-D) with a suitable/optimal gap distance selected (that can be easily fabricated)? I feel the inclusion of this data in the abstract will help readers get a summary of the predicted device characteristics that are provided in Figure 2. Additionally, when experimenting with heat current switching, what is the reason for selection for a particular set of temperature and a gap size?

The source and drain electrodes are at room temperature, whereas the gate electrode's temperature varies from room temperature to ~400 K. The gap size in our experiments was chosen to maximize the switching ratio. Further, the temperature change of the gate electrode (~117 °C) was chosen such that the VO_x film undergoes a complete phase transition. This is now stated in the manuscript on page 4 of the manuscript, where we state:

“the silicon resistor is used to heat up the temperature of the gate region, which enabled us to systematically vary the temperature of the gate from 298 to 390 K to ensure a complete phase transition and thus control the phase (metal or insulator) of the VO_x film integrated into the gate.”

(5) The authors allude to the hysteretic behaviour associated with the phase transition properties of VO_x. Such behaviour can pose challenges if the device is intended for switching applications, potentially leading to functionality issues. Could the authors offer insights into any dead band limits associated with changes in source-drain conductance during the cycling of the transistor between its ON and OFF states in the experiment? Moreover, what implications does this behaviour have on the ON/OFF ratio?

Furthermore, it would be beneficial to understand how the authors accounted for noise threshold values, such as signal-to-noise ratio, in characterizing the switching characteristics. Given the inherent complexities of hysteretic behaviour, elucidating its impact on the device's performance and reliability is crucial for assessing its practical applicability.

We thank the referee for these comments. We note that the hysteretic behavior during the heating and cooling of the VO_x film, as reported in Fig. 3d of the manuscript and Fig. S3 of the SI, has no impact on the ON and OFF states of the thermal transistor. This is because the dielectric properties of the VO_x film are quite insensitive to the small differences in the resistance of the films that arise upon cycling (see Fig.

S3 of the SI). This insensitivity implies that the ON/OFF ratios are robust to these small changes. As explained in the response to comment 6 (see below), the noise floor of our measurements is ~ 5 pW, and the signals are ~ 250 pW or larger. Therefore, the signal-to-noise ratio is >50 in all our measurements, making our conclusions quite robust.

(6) Could the authors expand Figure 3 (d) or provide a comment on the thermal sensitivity of the device with respect to the variation of the heat current rates for the small variation in the temperatures?

We interpret this question to mean, what the change in the source-drain thermal conductance is if there is a small temperature change (~ 1 mK) in the gate electrode when the gate electrode is in the metallic or insulating states. We note that when VO_x has fully transitioned to a metallic or insulating phase, there is very little impact of small temperature fluctuations on the dielectric properties of VO_x . Therefore, there is no detectable change in thermal conductance for such small fluctuations. To provide further information, we note that the smallest heat current change we can detect in our drain electrode is ~ 5 pW, given the bandwidth of our measurements. See past work by some of us ⁷, where the resolution of similar calorimetric devices is discussed.

(7) The authors have not adequately portrayed the current state of thermal transistor technology in their work. The theoretical advancements in this field have progressed rapidly, encompassing steady-state model [4], periodic drive analyses [5], and stochastic small signal model [6]. It would be beneficial to include a discussion of these advances in the paper, as they provide insights into the physical possibilities for implementation.

[4] Bao-qing Guo et al, Multifunctional quantum thermal device utilizing three qubits, *PhysRevE*.99.032112.

[5] Nikhil Gupt et al. "Floquet quantum thermal transistor," *Phys. Rev. E* 106, 024110 (2022)

[6] Uthpala N. Ekanayake et al., "Stochastic model of noise for a quantum thermal transistor," *Phys. Rev. B* 108, (2023)

For completeness, we now add references to these papers on page 2, where we now state.

"More recently, several theoretical works have also explored the implementation of thermal transistors via quantum effects^{20-23,}"

Report of Referee C

The authors present a compelling study on a nanoscale radiative thermal transistor utilizing a gate structure. They demonstrate control over radiative heat transfer between source and drain via insulator-metal phase transition in the material, significantly impacting guided modes. An on/off ratio of ~ 3 is achieved at a gap size of $\sim 1 \mu\text{m}$ with sub-second switching time.

The experimental setup exhibits high reliability with meticulous system and measurement calibration. The manuscript is clear, well-written, and supported by sound experimental data and simulations. Given the current interest in thermal transistors within the heat transfer community, this timely demonstration will undoubtedly generate significant attention.

For the reasons stated above, I recommend this manuscript for publication after minor revision.

We thank the referee for thoughtful feedback and for recommending publication after minor revisions. Below, we address all the questions raised by the referee.

(1) Total Power Calculation (Fig. 4): Supplement the spectral conductance data with calculations of total power as a function of gap distance. Compare this data with the measurement results in Fig. 2d. The simulation should also include a comparison of the total power ratio between insulating and metallic phases, along with the gap size where conductance drops (observed at $10 \mu\text{m}$ in the experiment).

The spectral plots obtained in Fig. 4 are generated from the same SCUFF-EM model that was used to generate Fig. 2d, which provides the thermal conductance as a function of gap size (the power is simply the conductance multiplied by the temperature differential). Below, we provide the power for a temperature differential of $\sim 4.9 \text{ K}$, which corresponds to the applied temperature differential in the experiments (this is now added on page 11 of the SI). As can be seen, the experiments and the simulation agree qualitatively well with each other, but they quantitatively differ by about 30–40%. The differences are likely due to ignoring the contributions of supporting beams and uncertainty in the dielectric functions, as stated on page 11 of the manuscript. The power ratio between the metallic and insulating phases is provided in Fig. 3b.

Supplementary Figure S9. Comparison of total power radiated from the source to the drain between the experiments and simulations. a, Total radiative power vs gap 'd' for the case when the gate is metallic, b, same as in 'a' but the gate is in its insulating phase.

(2) Comparison Table (Page 9): For clarity, create a table summarizing on/off ratios and switching times for easier comparison with previous works mentioned on page 9.

We thank the referee for providing this suggestion. We have now revised the manuscript on page 9 and added a comparison table on page 24 of the manuscript.

On page 10:

“To compare the performance of our thermal transistor with the other thermal transistors and switches, we summarize results from multiple works in Table 1.”

On page 23:

Table 1. Summary of some of the thermal transistor and thermal switch characteristics from past and current works.

Study	Gating mechanism	Operating temperature	Switching ratio	Switching time
J. Cho et al. , Ref. [41]	Electrochemical	Room Temperature	~1.5	~1.5 h
A. Sood et al. , Ref. [26]	Electrochemical	Room Temperature	~10	~7 min
Q. Yang et al. , Ref. [24]	Electrochemical	280 °C	~4	~ 3 min
M. Li et al. , Ref. [47]	Electric Field	Room Temperature	~13	~1 μ s
L. Castelli et al. , Ref. [25]	Thermal	Room Temperature	109 \pm 44	$\tau_{\text{OFF-ON}} = 1.7$ min $\tau_{\text{ON-OFF}} = 4.3$ min
Y. Li et al. , Ref. [27]	Thermal	Room Temperature	14.3	—
Current work	Thermal	Room Temperature	~3	$\tau_{\text{OFF-ON}} = \sim 470$ ms $\tau_{\text{ON-OFF}} = \sim 500$ ms

(3) *Discrepancy Explanation (Fig. 3b): Address the noticeable discrepancy between experimental data and simulations around the 5 μ m gap size in Fig. 3b. Provide an explanation for this difference.*

This discrepancy is related to the data shown in Figs. 2c and 2d where we find that the source drain conductance ($G_{\text{S-D}}$) in our experiments is higher for the metallic phase of VO_x than for the insulating phase for all gap sizes, including intermediate gaps. However, the model results (where we ignore the beams suspending the Source and the Drain) do not show a significant difference in $G_{\text{S-D}}$ for the two phases when the gap size is around 5 μ m. This discrepancy results in the disagreement between the experimental and simulated data in Fig. 3b. As alluded to above, we believe that this discrepancy possibly arises from the simplifications in the model geometry and, to a lesser degree, due to the uncertainty in the dielectric functions. We now acknowledge this difference on page 8 of the manuscript, where we state:

“The noticeable disagreement between the experimental and simulated data in Fig. 3b for gap sizes of around 5 μ m likely arises from the simplifications in the model geometry used in our simulation and, to a lesser degree, from the uncertainty in the dielectric functions.”

(4) Oscillation Source (Fig. 3f): Explain the source of the periodic oscillation observed in Fig. 3f.

We measured the time response (Fig. 3f) of our thermal transistor using a lock-in amplifier. Specifically, for obtaining the data shown in Fig. 3f we used a very small time constant (100 ms) of the lock-in to capture the high switching speed of our transistor device. This choice of a small time constant, while critical for observing the switching process, results in sinusoidal oscillations of the output of the lock-in amplifier.

(5) Missing Axis Labels (Figs. 4c-f): Include missing axis labels in Figs. 4c-f.

Thank you for your feedback regarding the missing axis labels in Figs. 4c-f. We have now added the axis labels to the revised figure in the manuscript.

(6) Terminology: It would be better not to use the term “novel” in the abstract and summary.

We have now deleted the term 'novel' from both the abstract and summary.

References

1. Zhou, Y. *et al.* Voltage-triggered ultrafast phase transition in vanadium dioxide switches. *IEEE Electron Device Lett.* **34**, 220-222 (2013).
2. Cavalleri, A. *et al.* Femtosecond structural dynamics in VO₂ during an ultrafast solid-solid phase transition. *Phys. Rev. Lett.* **87**, 237401 (2001).
3. Kumar, C.V.S., Maury, F. & Bahlawane, N. Vanadium Oxide as a Key Constituent in Reconfigurable Metamaterials. *Metamaterials Metasurfaces* (2018).
4. Wang, D. *et al.* in 2009 4th IEEE International Conference on Nano/Micro Engineered and Molecular Systems 593-596 (IEEE, 2009).
5. Ma, Y. *et al.* Full-scale simulation and experimental verification of the phase-transition temperature of a VO₂ nanofilm as smart window materials. *Mater. Today Commun.* **35**, 105758 (2023).
6. Li, M. *et al.* Electrically gated molecular thermal switch. *Science* **382**, 585-589 (2023).
7. Sadat, S., Meyhofer, E. & Reddy, P. Resistance thermometry-based picowatt-resolution heat-flow calorimeter. *Appl. Phys. Lett.* **102** (2013).

REVIEWERS' COMMENTS

Reviewer #1 (Remarks to the Author):

The author has answered all the questions I concerned, I recommend publishing this article.

Reviewer #2 (Remarks to the Author):

As the authors have now adequately addressed the comments, and the clear results and comparisons discussed in the revised manuscript shows promise in advancing the field of thermal management, the article is recommended for publication.

Reviewer #3 (Remarks to the Author):

The authors responded to all questions and revised the manuscript accordingly. The added Table 1 clearly shows the novelty of this paper. I recommend publication in this journal.